# Current Status of the Degradation of Aliphatic and Aromatic Petroleum Hydrocarbons by Thermophilic Microbes and Future Perspectives

**DOI:** 10.3390/ijerph15122782

**Published:** 2018-12-07

**Authors:** Alexis Nzila

**Affiliations:** Department of Life Sciences, King Fahd University of Petroleum and Minerals Dhahran, PO Box 468, Dhahran 31261, Saudi Arabia; alexisnzila@kfupm.edu.sa; Tel.: +96-613-860-7716

**Keywords:** biodegradation, thermophiles, petroleum hydrocarbons, aliphatics, aromatics, metabolites

## Abstract

Contamination of the environment by petroleum products is a growing concern worldwide, and strategies to remove these contaminants have been evaluated. One of these strategies is biodegradation, which consists of the use of microorganisms. Biodegradation is significantly improved by increasing the temperature of the medium, thus, the use of thermophiles, microbes that thrive in high-temperature environments, will render this process more efficient. For instance, various thermophilic enzymes have been used in industrial biotechnology because of their unique catalytic properties. Biodegradation has been extensively studied in the context of mesophilic microbes, and the mechanisms of biodegradation of aliphatic and aromatic petroleum hydrocarbons have been elucidated. However, in comparison, little work has been carried out on the biodegradation of petroleum hydrocarbons by thermophiles. In this paper, a detailed review of the degradation of petroleum hydrocarbons (both aliphatic and aromatic) by thermophiles was carried out. This work has identified the characteristics of thermophiles, and unraveled specific catabolic pathways of petroleum products that are only found with thermophiles. Gaps that limit our understanding of the activity of these microbes have also been highlighted, and, finally, different strategies that can be used to improve the efficiency of degradation of petroleum hydrocarbons by thermophiles were proposed.

## 1. Introduction

Contamination with petroleum hydrocarbons, as the result of oil exploitation, transport and refinement, is a growing concern worldwide. Thus, their removal from the environment remains a priority. Several approaches exist. However, biodegradation, which exploits the ability of microorganisms to use pollutants as a source of carbon and energy, remains the most attractive, since it is relatively cheaper and environmentally friendlier than chemical and physical approaches.

Petroleum hydrocarbon consists of alkanes (aliphatic compounds with up to 40 or more carbon [C] atoms); monocyclic aromatic hydrocarbons (MAHs, such as benzene, phenol, and their derivatives); polycyclic aromatic hydrocarbons (PAHs), which are subdivided further into low- and high-molecular weight PAHs (LMW-PAHs and HMW-PAHs, respectively). LMW-PAHs consist of 2–3 rings, such as naphthalene (NAPH), phenanthrene (PHEN), anthracene (ANTH), and fluorene (FLR), while HMW-PAHs contain four or more rings, and these include fluoranthene (FLT), pyrene (PYR), benzo[*a*]pyrene (BZP) and benzo[*a*]anthracene (BZA) [1,2].

Generally, aliphatic compounds are more amenable to biodegradation than aromatic compounds, and the higher the number of rings, the more difficult is the biodegradation. Nevertheless, the literature is replete with reports on the biodegradation of petroleum hydrocarbons, including HMW-PAHs, in mesophilic conditions. Mesophiles (bacteria that grow optimally at 20–40 °C) that degrade petroleum hydrocarbons have been extensively studied (primarily in low salinity, <4% NaCl, and at neutral pH), and are estimated to belong to more than 65 genera [3], and readers are referred to the following excellent reviews on this topic [2,4,5,6,7,8].

However, contaminations can also occur in extreme environmental conditions, and microorganisms that thrive in these environments are known as extremophiles. These conditions include high salinity (for halophilic microbes), high or low pH (alkalophilic or acidophilic microbes, respectively), high or low temperature (psychrophilic and thermophilic microbes, respectively), and high pressure (barophilic or piezophilic microbes) [9]. In relation to the biodegradation of petroleum hydrocarbons, a sizable body of work has been carried out in hypersaline conditions, and this has been summarized elsewhere [10,11,12].

Thermophiles fall into three groups. The moderate and extreme thermophiles, with optimum biodegradation temperatures of 40–59 °C and 60–85 °C, respectively, and the hyper-thermophiles, with optimum temperatures >85 °C [13,14]. Several advantages are associated with the use of thermophiles (Figure 1). High temperatures lead to increased kinetic reactions (thus, high reaction rates) and high mass transfer, thus increasing the biodegradation. In addition, high temperature has a significant influence on the bioavailability of petroleum hydrocarbons, especially PAHs, by decreasing their viscosity, leading to an increase in their diffusion coefficients, therefore improving their availability to microorganisms [13,14]. All of these properties have made thermophiles a valuable source of enzymes in industrial biotechnology [15,16,17].

As stated earlier, biodegradation of petroleum hydrocarbon has focused on mesophiles, and comparatively little work has been carried out on petroleum hydrocarbon degrading thermophiles (PHDT). Several reviews on PHDT have been published in the last 20 years [18,19,20,21]; however, these reviews lack detailed analyses of the characteristics of PHDT, the mechanisms of degradation, and also comparative information between thermophiles and mesophiles in relation to petroleum product degradation. In the current review, work carried out on this topic since the 1980s was reviewed, and detailed observations of this work (in comparison to that of mesophiles) led to the unravelling of important and unique features of thermophiles involved in the degradation of petroleum hydrocarbons, which have not been described so far. For instance, more than 75% of PHDT belong to one single genus, *Geobacillus*, a situation different from that which prevails in mesophiles. Unique catabolic pathways (present only in PHDT) were highlighted. More importantly, gaps that limit our understanding of the activity of these microbes have also been presented, and, finally, different strategies that can be used to improve the efficiency of PHDT were proposed. Overall, this review opens up new research avenues on this topic.

## 2. Summary of Work on PHDT

In 2001, Nazina et al. re-examined the genus *Bacillus* and proposed a new genus, *Geobacillus*, for some bacteria that were initially classified as *Bacillus*. Specifically, those authors proposed that species such as *Bacillus stearothermophilus*, *Bacillus thermoleovorans* and *Bacillus thermodenitrificans* should be classified in the genus *Geobacillus*, as the renamed species *Geobacillus stearothermophilus*, *Geobacillus thermoleovorans*, and *Geobacillus thermodenitrificans* [22]. Thus, to be in line with this new proposed classification, in this review, the aforementioned *Bacillus* species were named *Geobacillus* species even if they were reported before 2001.

### 2.1. Alkane Biodegradation

The first report on PHDT dates back to the mid-1960s. Mateles et al. isolated a thermophile, *Geobacillus stearothermophilus*, from soil water/soil samples from the US, by enrichment in a medium containing hydrocarbon at 57 °C (Table 1, ST1) [23]. Almost a decade later, an enrichment experiment on mud samples (from an American littoral zone) at 50 °C, in the presence of n-hexadecane, led to the isolation of a thermophilic microbe, *Thermomicrobium fosteri* [24] (Table 1, ST2). Further studies showed that this bacterium could utilize n-alkanes (C_10_–C_20_), alkenes (C_14_–C_18_), alcohols (C_12_–C_17_), and ketones (C_14_–C_17_). However, compounds of <C_9_ (n-alkanes, alkenes and ketones) and <C_11_ (alcohols) were not utilized by this strain [24]. Two years later, a similar study, based on enrichment with heptadecane, led to the isolation of two thermophilic bacterial strains, YS-3 and YS-4, from an inoculum consisting of mud samples from Yellowstone National Park (Table 1, ST3). These strains could degrade C_10_–C_20_ n-alkanes and C_6_–C_8_ alkenes at an optimum temperature of 60 °C, but not <C_9_ (alkanes) nor <C_6_ (alkenes) [25]. Further investigations showed that YS-3 and YS-4 belong to the species *Thermoleophilum minutum* and *Thermoleophilum album*, respectively [26]. The inability of these strains to utilize shorter chain n-alkanes will be discussed in Section 3.2. Using a sample from a hot water spring on Kunashir Island, Russia, Loginova et al. isolated two thermophilic bacterial strains, *G. stearothermophilus* and *Thermus ruber*, capable of growing in the presence of paraffins at an optimum temperature of 60 °C [27] (ST4). A few years later, in 1984, using an inoculum consisting of mud and water from various sites across the USA, a thermophilic strain, *T. album*, was isolated by an enrichment experiment at 60 °C, in the presence of n-heptadecane. This strain was capable of degrading C_13_–C_20_ n-alkanes [28] (ST5). Three years later (ST6), the same research group, using the same methodology and the same inoculum, reported the isolation of a thermophilic bacterium *G. thermoleovorans* that degraded C_13_–C_20_ n-alkanes within the temperature range of 55–65 °C. Interestingly, this bacterium could not grow in the presence of n-alkanes with <C_9_ and >C_20_ [29].

Two strains of *G. stearothermophilus* (KTCC-B2M and KTCC-B7S) were isolated by enrichment of oil-contaminated samples from Kuwait, in the presence of crude oil. The two strains could degrade C_15_–C_18_ n-alkanes, while n-alkanes with lower carbon numbers, alkenes and aromatics were less efficiently degraded [30] (***ST7***). In another study, two anaerobic facultative strains, *G. thermoleovorans* B23 and H41, were isolated from produced water from deep petroleum reservoirs in Japan. These strains could degrade C_13_–C_26_ n-alkanes within a temperature range of 50–80 °C. However, less degradation was observed when shorter alkane chains (<C_12_) were used [31] (ST8). Further investigations demonstrated that these strains could degrade n-alkanes through terminal oxidation, followed by β-oxidation [31].

A strain NG80-2 of *G. thermodenitrificans* was isolated following an enrichment culture of a sample from a deep oil reservoir in China, in the presence of crude oil, at 73 °C. This strain could degrade C_15_–C_36_ n-alkanes, at an optimum temperature of 65 °C. However, no growth was observed with short-chain n-alkanes (C_8_–C_14_), and those >C_40_. Interestingly, as will be discussed in Section 3.3.1, *alkB*, a gene encoding the enzyme system responsible for the first step of the degradation of n-alkanes, was not found in this strain [32] (ST9); instead, a novel alkane monooxygenase, *LadA*, was identified [33]. In a different study, various species of *Geobacillus* (*G. thermoleovorans* strain 27, *Geobacillus caldoxylosilyticus* 17, *Geobacillus pallidus* 2, *Geobacillus toebii* 1, *Geobacillus* sp. 3) that degrade n-hexadecane have been isolated following an enrichment of contaminated soil samples. In that study, evidence indicated that the expression of *alkB* gene was induced by n-hexadecane [34] (ST10). The use of the same substrate, n-hexadecane, in the enrichment of a medium containing an inoculum from Dongxin oil reservoir, China, led to the isolation of a thermophilic bacterium TH2, that was able to degrade petroleum hydrocarbons, mainly n-alkanes, at an optimum temperature of 70 °C [35] (ST11). In that study, species identification was not carried out.

Recently, Tourova et al. reported three strains, *G. toebii* B-1024, *Geobacillus* sp. 1017 and *Aeribacillus pallidus* 8m3, capable of degrading C_10_–C_30_ n-alkanes [36] (ST12). The *alkB* gene, encoding rubredoxin-dependent alkane monooxygenase, was found in these three strains, while another gene, *ladA* gene, which codes for flavin-dependent alkane monooxygenase, was only present in the strains B-1024 and 1017 [36]. That study revealed (as will be discussed in Section 3.3.1), for the first time, the simultaneous presence of *alkB* and *ladA* genes, responsible for oxidation of medium-chain and long-chain n-alkanes [36]. In addition, the genome of *Geobacillus* sp. 1017 was sequenced, and several key genes associated with the degradation of petroleum hydrocarbons have been identified [37].

In the aforementioned studies, bacteria were isolated following enrichment in the presence of hydrocarbons. However, in 2006, hydrocarbon degrading bacteria were isolated using an *alkJ* probe, an alkane hydroxylase gene involved in the degradation of n-alkanes [38] (ST13). Bacteria in volcanic samples from Greece were first cultured in rich media in a temperature range of 60–80 °C, and around 150 strains were isolated. Thereafter, genetic investigations using the *alkJ* probe led to the identification of 10 strains, and four were *G. thermoleovorans*, three *G. stearothermophilus*, two *Geobacillus anatolicus* and one *Bacillus aeolius* [38]. Further studies confirmed that these strains could utilize long-chain n-alkane compounds from crude oil [38]. As will be discussed in Section 4.1.2, the use of genetic approaches provides an alternative to the selection and isolation of PHDT.

### 2.2. Monocyclic Aromatic Hydrocarbons (MAHs)

The first evidence of MAH biodegradation was provided in 1974. A strain of *G. stearothermophilus* PH24 was selected from industrial sediment and grew at 55 °C in the presence of phenol. Further studies indicated that, in addition to phenol, this strain could also degrade catechol, at an optimum temperature of 50 °C [39] (ST14). Evidence was also provided that this strain could cleave catechol at the meta-position. A year later, the same strain was shown to degrade other MAHs including o-, m- and p-cresol, 3-methylcatechol and 4-methylcatechol [40,41] (ST15).

Using soil that had been pasteurized for 10 min at 80 °C, Adams and Ribbons isolated a strain of *G. stearothermophilus* IC3 by enrichment in the presence of phenol and m-cresol. In addition to m-cresol and phenol, this strain could also utilize benzoic acid, and the degradation of phenol led to catechol, followed by cleavage at the meta-position [42] (ST16). A year later, another phenol-degrading *G. stearothermophilus* strain BR219 was isolated (by enrichment in the presence of phenol), using river sediment from the USA (ST17) [43]. Further investigation led to the characterization of phenol hydroxylase, the first enzyme involved in phenol degradation [44].

Dong et al. (1992) reported the isolation of a *G. stearothermophilus* strain FDTP-3; however, detailed information was not available on the source of the inoculum and the conditions of the isolation [45] (ST18). The strain could degrade phenol along with catechol, but not benzoic acid. Two genes of the meta pathway of phenol degradation, phenol hydroxylase and catechol 2,3-dioxygenase, were also characterized by cloning in *Escherichia coli* [45,46]. Using phenol as the sole substrate, a *Geobacillus* sp. A2 strain was isolated from a hot spring sample (from Iceland), in an enrichment experiment at 70 °C [46,47] (*ST19*). This strain could grow efficiently in the presence of phenol or isomers of cresol (o-, m-, and p-cresol) at 70 °C. In addition, the cleavage of the aromatic ring proceeds through the meta-position. Genetic analysis showed that this A2 strain was closely related to *G. thermoleovorans* [46,47]. In a different study, two strains, *Thermus aquaticus* ATCC 25,104 and *Thermus* sp. ATCC 27978, were shown to degrade the MAHs benzene, toluene, ethylbenzene, and xylene, at 70 °C (for strain ATCC 25104) and at 60 °C (for ATCC 27978), respectively [48] (ST20). In addition, using ^I4^C-labeled benzene and toluene, radioactive ^14^CO_2_ was identified, illustrating the complete mineralization of these compounds [48]. In another study, three strains of phenol degrading *Bacillus* were isolated following an enrichment of contaminated samples in the presence of phenol, at 50 °C [49] (ST21). These strains could degrade phenol (along with o-, m-, and p-cresol) in an optimum temperature range of 50–55 °C, and this degradation (for phenol) is carried out via the meta-pathway. The presence of catechol 2,3-dioxygenase was also detected in this study [49].

A halotolerant and thermophilic strain, *Aeribacillus pallidus* VP3, was isolated following enrichment of an inoculum consisting of production water from a Tunisian oil field, in the presence of the MAH vanillic acid, at 55 °C. The selected strain VP3 could degrade several MAH compounds including benzoic, p-hydroxybenzoic, protocatechuic, gallic, p-coumaric, ferulic and caffeic acids [50] (ST22).

All of the aforementioned studies were carried out in bacteria. However, archaea have also been shown to degrade MAHs. For instance, a thermophilic strain of *Sulfolobus solfataricus* P2 isolated from a volcanic site, was sequenced, and its genome revealed the presence of genes encoding enzymes involved in MAH degradation, including mono- and dioxygenases [51]. This information led Izzo et al. to investigate the ability of this archaean strain to grow in the presence of MAH [52] (ST23). The results indicated that *S. solfataricus* P2 could actively grow on phenol as the sole source of carbon, at an optimum temperature of 80 °C, and the presence of phenol induced the expression of enzymes involved in the degradation of MAHs. As will be discussed in Section 4.1.2, this study illustrates the usefulness of the exploitation of genome information to identify potential substrates (pollutants) that a given microorganism can degrade [52].

### 2.3. Polycyclic Aromatic Hydrocarbons (PAHs)

In 1999, Shimura et al. provided the first evidence of the degradation of a PAH by thermophiles. The authors carried out an enrichment culture of a compost from a field in Okayama, Japan, at 60 °C using biphenyl as the sole source of carbon, leading to the isolation of *Geobacillus* sp JF8 strain [53] (ST24). This strain could degrade biphenyl, naphthalene, and the MAH benzoic and salicylic acids. Polychlorinated biphenyl congeners including tetra- and penta-chlorobiphenyl could also be degraded. However, no degradation was observed with the aromatic compounds PHEN, ANTH, benzene, o-xylene, m-xylene, p-xylene and toluene [53].

The ability of two thermophilic strains, *Bacillus subtilis* BUM and *Mycobacterium vanbaalenii* BU42, degrading PHEN were tested in degrading BZP at 55 °C. Neither of the strains could degrade BZP when used as the sole source of carbon [54]. However, when BZP was used as co-substrate with PHEN, the strain BUM could degrade BZP, while the strain BU42 could not [54] (ST25). This investigation illustrated the usefulness of cometabolism (in the case of the BUM strain) in the removal of recalcitrant PAH, as will be discussed in Section 4.2.2.

Another example of HMW-PAH degradation by cometabolism was provided by Feitkenhauer and Markl [55] (ST26). Using a mixture of thermophilic bacteria *Bacillus* sp. and *Thermus* sp. pre-isolated from hot springs, compost piles and industrial wastewater, the authors showed that these two strains could degrade the PAHs acenaphthene, FLT, PYR and BZP, but only in the presence of hexadecane [55] (ST26). In this context of cometabolism, hexadecane was the growth substrate, while PAHs were non-essential substrates, as will be discussed in Section 4.2.2.

Using soil samples from contaminated areas in Kuwait as inoculum, several thermophilic bacteria were isolated by enrichment in the presence of a mixture of the following PAHs (NAPH, PHEN, ANTH, PYR and FLR) and the heterocyclic polyaromatics (benzothiophene, dibenzothiophene dibenzofuran and carbazole), at 60 °C. Several thermophilic bacteria were isolated, including *Bacillus firmus*, *Bacillus pallidus*, *Anoxybacillus* sp., *Paenibacillus* sp., and *Saccharococcus* sp. [56] (ST27). Interestingly, when glucose was used as substrate, the thermophilic bacteria growth was inhibited, and this inhibition was explained by catabolite repression, in which glucose acts as a repressor for the synthesis of some enzymes involved in the biodegradation of PAHs [56]. The existence of this inhibitory effect in the context of cometabolism during degradation of petroleum hydrocarbons has already been reported [57], and as will be discussed in Section 4.2.2, this inhibition puts a caveat on the process of cometabolism.

Three thermophilic *Geobacillus* sp. strains have been isolated following an enrichment of compost in the presence of PHEN at 60 °C. Further studies have shown that these strains could also degrade the PAHs FLR and FLT [58] (ST28). A thermophilic bacterium, *Nocardia otitidiscaviarum* TSH1, was isolated following an enrichment of petro-industrial wastewater soil from Iran, in the presence of NAPH. The strain could degrade NAPH at 60 °C, and further analyses have led to the identification of key NAPH metabolites [59,60] (ST29). As will be discussed in Section 3.3.3, overall, the NAPH degradation pathway differs from that of mesophilic bacteria. The same strain has also been shown to degrade PHEN and ANTH, and several of their metabolites have been identified [60] (Section 3.3.3). In another study, an NAPH degrading thermophilic bacterium, *Geobacillus* sp. G27, was isolated from a geothermal oil field in Lithuania. The strain could degrade NAPH, along with ANTH, protocatechuic acid, benzene-1,3-diol, phenol and benzene at an optimum temperature of 60 °C. However, no growth was observed with catechol [61] (ST30). Chromatographic analyses have indicated that the first NAPH ring opening occurred via ortho-cleavage, leading to protocatechuic acid, as will be discussed in Section 3.3.3.

In 2000, the enrichment of a hydrocarbon contaminated soil from Germany in the presence of NAPH at 60 °C led to the isolation of a *G. thermoleovorans* Hamburg 2 strain [62] (ST31). Further studies showed the presence of some NAPH metabolites generated by this strain, such as benzoic acid, which has not been commonly reported in mesophiles (Section 3.3.3).

Two strains of *Geobacillus* genus, *G. pallidus* XS2 and XS3, isolated from oil contaminated soil from China, were reported to degrade PHEN and FLR. The strains were selected following enrichment in the presence of PHEN, FLR and crude oil, at 60 °C. Further analyses showed that more than 70% of the tested PAHs were degraded within 20 days (from 250 mg/L), and, in addition, the strain could degrade n-alkanes and produce bioemulsifier [63] (ST32). In another study, the enrichment of a soil sample from a Chinese oilfield, in the presence of the crude oil at 70 °C, led to the isolation of *Geobacillus* sp. SH-1. This strain could degrade NAPH and C_12_–C_23_ n-alkanes [64] (ST33). Several metabolites of NAPH have been identified (Section 3.3.3).

Recently, *G. stearothermophilus* strain A-2 was isolated from produced water from Dagang petroleum reservoir, China, at a temperature of 73 °C. Further studies showed that this strain could degrade NAPH, methylated PHEN, FLR, benzo[*b*]fluorenes and long-chain alkanes (nC_22_–nC_33_). However, shorter chains (nC_14_–nC_21_) were not efficiently degraded. In addition, this strain had a strong surface hydrophobicity and produced a bioemulsifier, making it an ideal strain for bioremediation [65] (ST34). In another recent study, a strain of *Bacillus licheniformis* M2-7 was isolated from an inoculum from a hot spring in Mexico, and was proved to degrade BZP. Further studies have led to the identification of the MAH phthalic acid as a BZP metabolite, and the characterization of the catechol dioxygenase enzyme, a clear illustration of the ability of this strain to convert BZP to aliphatic derivatives [66] (ST35).

## 3. General Findings

### 3.1. Geobacillus Are the Most Dominant PHDT

As Table 1 shows, a total of 49 different strains of thermophilic bacteria and one archaea (*S. solfataricus*) degrading hydrocarbons were isolated and/or characterized, and they all fall in the following 12 genera (Figure 2): *Thermomicrobium*, *Sulfolobus*, *Anoxybacillus*, *Paenibacillus*, *Saccharococcus*, *Mycobacterium* and *Nocardia* at 1.7% each, *Aeribacillus* (3%), *Thermoleophilum* (5%), and *Bacillus* (10%); the most striking observation was that the bulk of strains, around 63%, belong to the genus *Geobacillus*.

It is even more surprising that these strains were selected in different parts of the world (encompassing the five continents and around 20 different countries) and from different sources including hot springs, produced water, soil samples, geothermal oil fields, and compost samples, among others. Thermophiles are known to be distributed in more than 70 genera, with a total of 140 species [35]. Although only seven genera of these thermophiles have been reported in this review, however, the skew of species to *Geobacillus* genus deserves further discussion.

In comparison, the distribution of the mesophiles that degrade hydrocarbons is wider. More than 150 different genera have been reported, and a careful observation of the seminal reviews on this topic clearly shows that not a single genus represents even more than 25% of the reported species [4,5,7,67]. This begs the question, why do thermophiles degrading hydrocarbons belong predominantly to the genus *Geobacillus*?

Bacteria of *Geobacillus* genus are endospore-forming microbes, and are aerobic or facultative anaerobic. They are primarily thermophiles, thus are found to proliferate in environments where temperatures are relatively high. However, surprisingly, they have even been found in a wide range of moderate and low-temperature environments, including the deep sea, which are below their minimum requirement for growth [17,68]. Evidence indicates that their endospores resist high stress conditions, including temperature variation, UV light and desiccation [68]. These bacteria are also characterized by their lower size of spores (around 1 µm diameter), allowing them to remain suspended in the atmosphere for long periods, time that can be sufficient for their widespread distribution to all continents [17,68]. In addition, genomic investigations show that these microbes have a battery of gene encoding transporters and hydrolytic enzymes, specifically those adapted to using plant biomass as a source of carbon and energy [17,68,69,70]. These observations, coupled with the high reproduction rate of *Geobacillus*, have been proffered to be the driving factors for the ubiquitousness of this thermophilic genus on earth [17,68,69,70]. However, their high presence in environments contaminated with petroleum products underscores that these bacteria are also endowed with gene encoding enzymes involved in the degradation of both aliphatic and aromatic products [17], and as discussed in Section 3.3, some of these catabolic genes (or enzymes) have been characterized.

It is also worth noting that, among the *Geobacillus* genera, *G. stearothermophilus* and *G. thermoleovorans* represent 67% of all *Geobacillus* species. *G. stearothermophilus* is the most prominent *Geobacillus* species (30%), and is also the reference species in this genus [22]. Finally, almost 81% of the reported thermophiles (Table 1) belong to the family *Bacillaceae.*

### 3.2. Short Chain n-Alkanes Are Less Efficiency Degraded

Most studies in which several n-alkanes were investigated showed clearly that, in general, shorter chain n-alkanes (generally <C_9_) were less efficiently degraded than longer chains (>C_25_) [3,4,5,6,7,8,9] (ST2).

These observations are in line with those made on mesophiles, that midsized n-alkanes (C_10_–C_20_) are generally more amenable to degradation than shorter or longer chain n-alkanes. One of the reasons for the slow utilization of longer chains is their low water solubility, thus their low bioavailability compared to midsized n-alkanes [71]. On the other hand, although short-chain n-alkanes have higher aqueous solubility, they are toxic to microbes or cells, and the mechanisms of this toxicity are associated with their uptake and dissolution in the cell membrane [71]. Thus, though short chain n-alkanes are more bioavailable than midsized or long chain n-alkanes, they are associated with cell toxicity, hence their inhibition of microbial growth.

### 3.3. Specific Observations in Relation to Mechanisms of Degradation

#### 3.3.1. n-Alkanes

In general, the mechanisms of n-alkane degradation proceed in a similar fashion with both mesophiles and thermophiles. This process is initiated by the action of monooxygenases, enzymes that incorporate one atom of O_2_ into the n-alkane chain, or by dioxygenases, enzymes that can incorporate two O_2_ atoms. Both pathways will lead to the generation of alcohol, which will then be converted into fatty acids, and the latter will then be subjected to β-oxidation to generate energy and CO_2_ through the central metabolism of the bacteria [72]. The genes involved in this pathway have been studied in mesophiles, specifically in the *Pseudomonas* genus. The n-alkane pathway is controlled by two operons: The *alkBFGHJKL* operon encodes components of the alkB system, the membrane-bound n-alkane hydroxylase system, and the alkT/alkS operon. The latter, lkT/alkS operon, encodes rubredoxin reductase enzyme, which also regulates the expression of the *alkBFGHJKL* operon [73,74]. This *AlkB* gene has also been characterized in thermophilic bacteria *Geobacillus* and *Aeribacillus* [34,36] (ST10, 12). Thus, it is likely that other genes belonging to these operons are present in thermophiles that degrade petroleum products. For instance, as will be discussed in Section 4.1.2, the *alkJ* gene from *Pseudomonas* has been used to detect and isolate n-alkanes utilizing thermophilic strains of *G. thermoleovorans, G. stearothermophilus, G. anatolicus* and *B. aeolius* (ST13).

In addition to the AlkB system, two other oxygenases have been reported in mesophiles. These are cytochrome P450 monooxygenases, which belong to the cytochrome CYP153 family [75], and flavin-dependent dioxygenases, encoded by the *almA* gene [76]. AlkB and CYP153 enzymes are generally implicated in the oxidation of medium-chain C_5_–C_17_ n-alkanes while the almA system is associated with the oxidation of long-chain C_10_–C_30_ n-alkanes [73,75,76].

Interestingly, so far, the two systems, CYP153 and AlmA, have not been reported in thermophilic bacteria degrading n-alkanes. Instead, these thermophiles express a novel type of alkane monooxygenase system, LadA, which was initially discovered in a *G. thermodenitrificans* NG80-2 strain [32,33] (ST9). Similar to AlmA, LadA is a flavin-dependent monooxygenase of the family of bacterial luciferases, and is responsible for the degradation of long-chain n-alkanes C_15_–C_36_. Recently, the simultaneous presence of *alkB* and *ladA* genes has been reported in two strains of *Geobacillus* and one of *Aeribacillus* [36] ([ST12).

In summary, so far, research has shown that n-alkane degrading mesophiles and thermophiles express AlkB monooxygenase enzymes; however, in addition, these mesophiles have CYP153 and AlmA enzymes, while these thermophiles express LadA monooxygenases.

#### 3.3.2. MAHs

Phenol has been the most studied MAH in thermophiles. As in mesophiles, the biodegradation of this compound is initiated by hydroxylation to generate catechol, a reaction mediated by phenol hydroxylase. In mesophiles, this catechol is metabolized either through ortho- or meta-cleavage (Figure 3), in the presence of catechol 2,3-dioxygenase enzyme. However, detailed observation of the data published for thermophiles has shown that, so far, only meta-cleavage is present in these thermophiles (ST14, 16, 19, 21) (Figure 3). The two enzymes, phenol hydroxylase and catechol 2,3-dioxygenase, have been characterized in *G. stearothermophilus* (ST17, 18) and *Bacillus* sp. (ST21). Using a *G. stearothermophilus* strain, Omokoko et al. reported the complete sequence of an operon encoding the meta-pathway genes, and characterized the first transcriptional regulator of phenol metabolism [77]. In summary, thermophiles differ from mesophiles in their preference for the meta-cleavage of MAHs.

#### 3.3.3. PAHs

The mechanism of PAH degradation in thermophiles has received little attention. So far, in total, only five reports have studied this. Four have been dedicated to NAPH (ST29, 30, 31, 33), and in one of them (ST29), the analyses were extended to PHEN and ANTH; and one study investigated the biotransformation of BZB (ST35).

Like mesophiles, thermophiles initiate the degradation of PAHs (in this case, NAPH, PHEN and ANTH) by the action of mono- or dioxygenases, enzymes that mediate the formation of mono- and then di-hydroxylated aromatic compounds. As Figure 4, Figure 5 and Figure 6 show, the mono- and di-hydroxylated derivatives of NAPH, PHEN and ANTH have been identified (ST29, 30, 31, 33). Thereafter, these di-hydroxy-aromatic compounds are subjected to cleavage, either between the hydroxyl groups (meta-cleavage) or proximal to one of the two hydroxyl groups (ortho-cleavage), to generate an aromatic compound with less than one ring. A careful observation of NAPH degradation (Figure 4) indicates that cleavage of the first ring occurs at the meta-position. For instance, the derivatives 3-(2-hydroxyphenyl)-propanoic acid (ST31), 2-hydroxycinnamic acid (ST29), 4-(2-hydroxyphenyl)-2-oxo-but-3-enoic acid (ST31, 33) and (2E)-3-(2-hydroxyphenyl)prop-2-enal (ST33) are likely to result from the ring opening of 1,2-dihydroxynaphthalene at the meta-position. Likewise, the existence of 3-(2-carboxyphenyl)-2-propenoic acid (2-carboxycinnamic acid) (ST29, 31) could result from the meta-cleavage of 2,3-dihydroxynaphthalene (Figure 4).

Study ST30 identified protocatechuic acid, and, so far, the existence of this metabolite has not yet been identified in mesophiles, although it has been proposed [78].

Three studies (ST29, 31, 33) out of four on NAPH degradation showed the presence of a benzoic acid metabolite. To the best of the knowledge of the author, in only one study has this compound been reported in the context of NAPH degradation in mesophiles [79]. This metabolite has also been reported in the context of ANTH degradation (in mesophiles), and was proposed to be derived from decarboxylation of phthalic acid [80]. Interestingly, phthalic acid has been reported in thermophiles (ST30, 31). Thus, the fact that most studies on NAPH degradation have identified benzoic acid highlights the preference of thermophiles for this reaction (decarboxylation of phthalic acid). This benzoic acid was also reported in the degradation of ANTH by the thermophilic strain *N. otitidiscaviarum* TSH1 (Figure 5, ST29).

In relation to PHEN (Figure 5), the metabolite 4-[1-hydroxy(2-naphthyl)]-2-oxobut-3-enoic acid could be derived from the first ring opening, at the meta-position, of 3,4-dihydroxyphenanthrene, while 2,2-biphenic acid could result from ortho-cleavage of 9,10-dihydroxyphenanthrene. However, the observation of ANTH degradation (Figure 6) indicates that the first ring opening is the result of the ortho-cleavage of 1,2-dihydroxy-anthracene (which is probably derived from 1,2-dihydroxy-1,2-dihydroanthracene) to generate 3-(2-carboxyvinyl) naphthalene-2-carboxylic acid, another identified metabolite (Figure 6, ST29).

In summary, in thermophiles, the first ring opening of NAPH occurs at the meta-position only, as is the case with phenol, and that of ANTH proceeds through ortho-cleavage, while both meta- and ortho-cleavage exist in PHEN degradation.

## 4. Gaps and Strategies to Improve Biodegradation by Thermophiles

### 4.1. Isolation of More Thermophilic Microbes

#### 4.1.1. Use of Experimental Set-Up

Table 1 shows that most strains were active within the range 40–80 °C, with optimum temperatures of <70 °C (most of them being around 60 °C). Only one strain, *S. solfataricus* P2, an archaea, had an optimum temperature >70 °C (optimum temperature being 80 °C) (ST23). As discussed in Section 1, these microbes fall into the group of moderate- and extreme-thermophiles. Thus, so far, no hyper-thermophiles (whose optimum temperatures are >85 °C) that degrade petroleum hydrocarbons have been described. The following reasons could explain the absence of the hyper-thermophiles.

First, most conditions of isolation of these thermophilic strains had involved the enrichment technique. In this approach, inocula or samples that are presumed to contain microbes are incubated at a fixed temperature, in the presence of the substrate of interest, and most of these enrichment experiments were carried out at temperatures <70 °C; in fact, 55 and 60 °C were the most commonly used temperatures. During the enrichment process, microbes that are favored are those whose growth is optimum at temperatures set in the experiments. Therefore, it is not surprising that most strains had optimum temperatures <70 °C and no hyper-thermophiles were selected during these experiments.

Thus, it is reasonable to argue that, so far, researchers have deliberately focused their work on moderate- and extreme-thermophiles. Working on hyper-thermophiles requires maintaining temperature of 85–120 °C throughout the experimental conditions, which can be a limitation in the context of routine cultures. This may explain why most enrichment experiments were carried out at temperatures <70 °C, even when samples that are likely to contain hyper-thermophiles were used as inocula (for instance, samples isolated near a volcanic site, in ST13). Thus, the degradation of petroleum hydrocarbons by hyper-thermophiles awaits exploration.

#### 4.1.2. Genomic Approach

Genetic approaches can also be used to isolate pollutant-degrading microbes in general, or PHDT in particular. This strategy employs molecular probes that are based on the sequences of key genes encoding catabolic enzymes in the pollutant degradation pathway. The advantage of this approach is that the probe can be tailored to target a specific enzyme, thus a specific pathway, therefore excluding microbes that do not have that particular pathway. As discussed in Section 3.3.1, this approach has been used to isolate PHDT. For instance, using *alkJ* gene encoding alkane hydroxylases from *Pseudomonas putida* and *Pseudomonas oleovorans* (two mesophiles), Meintanis et al. screened a pool of 150 thermophilic bacteria isolated from a volcanic site for the presence of *alkJ* gene. A total of 10 thermophilic strains (nine *Geobacillus* and one *Bacillus* genera) were identified, and further studies proved that they could all degrade long chain n-alkanes and crude oil [38] (ST13).

In mesophiles, this approach has also been used to identify bacteria that degrade the recalcitrant PAHs (especially pyrene) by using a probe based on the sequence of the aromatic ring hydroxylating dioxygenase enzyme (ARHDO), responsible for initial dihydroxylation [81,82,83]. As shown in Table 1, only a few thermophiles that degrade complex PAH compounds have been isolated so far. Thus, for instance, the use of ARHDO probes (based on sequences from mesophiles) could provide an opportunity to isolate and characterize more thermophiles that degrade PAHs.

PHDT can also be identified by exploiting the available information on the genome sequences of thermophiles. Specific genes encoding enzymes involved in petroleum hydrocarbon (or any pollutant) degradation can be sought in the genome sequences, and if these genes are present, in vitro studies can be initiated to assess the thermophile’s ability to grow in the presence of these pollutants. As discussed in Section 2.2 (ST23), this approach has successfully been tested, and led to the identification of the archaean strain *S. solfataricus* P2 as a MAH degrader. Thus, more PHDT can be identified by employing this strategy, since the genomes of a sizable number of thermophilic strains have now been sequenced (https://www.ncbi.nlm.nih.gov/genome). New “omics” techniques, such as functional metagenomic, can allow the identification of genes (thus their protein functions) present in genomes of microbial communities without the need of prior in vitro culture [84,85,86]. Using this approach, genes involved in the degradation of various pollutants, including PAHs, have been identified, cloned and functionally expressed [87]. Thus, the exploitation of this approach in the context of hydrocarbon degradation by thermopiles will immensely contribute in discovering novel and unique genes, which can then be used to isolate novel PHDT strains.

### 4.2. Improving Biodegradation

Several strategies to improve biodegradation of the pollutants have been evaluated in the context of mesophiles. The same strategies can be applied and can add value to the degradation of petroleum hydrocarbons by thermophiles. These strategies are summarized next.

#### 4.2.1. Bioaugmentation

This strategy is based on the addition of preselected and active microorganisms to a microbial community to enhance the ability of this microbial community to degrade pollutants. In the context of the use of mesophiles, this approach has been employed in the biodegradation of various recalcitrant and complex pollutants [88,89,90,91]. As Table 1 shows, many single thermophilic strains have been isolated and characterized as degraders of petroleum hydrocarbons. Thus, each of these strains could potentially be used in the context of bioaugmentation. However, so far, not a single bioaugmentation experiment has been reported using thermophiles for the degradation of petroleum hydrocarbons.

#### 4.2.2. Cometabolism

Cometabolism has been widely used in mesophilic conditions to increase pollutant degradation [5]. This approach uses two different substrates (pollutants), a non-growth and a growth-promoting substrate. As the name indicates, the non-growth substrate alone does not efficiently support microbial growth, leading to slow growth. However, when the growth-promoting substrate is added, microbial growth will increase, and this, in turn, will lead to degradation of the non-growth substrate [5].

Interestingly, three studies have evaluated this concept of cometabolism in the context of thermophiles (ST25, 26, 27). In the first (ST25), the strain *B. subtilis* BUM could not degrade the recalcitrant BZP when used as sole substrate. However, the addition of PHEN, a growth-promoting substrate, led to the degradation of BZP, a classic illustration of cometabolism. Likewise, in ST26, the degradation of the PAH compounds acenaphthene and FLT, PYR and BZP by a mixture of strains of *Geobacillus* sp. and *Thermus* sp. were augmented when the n-alkane hexadecane was used as a growth-promoting substrate.

However, as has been shown in mesophilic conditions, the use of growth-promoting substrates can inhibit the degradation of the non-growth substrate [92,93]. For instance, in mesophilic conditions, one study showed that the use of glucose (as growth-promoting substrate) inhibited the expression of enzymes involved in PAH biodegradation [93]. ST27 confirms this limitation in the context of thermophiles. Indeed, the use of glucose led to the inhibition of degradation of the PAHs NAPH, PHEN, ANTH, PYR and FLR, and the heterocyclic compounds benzothiophene, dibenzothiophene, dibenzofuran and carbazole. Thus, as discussed in a recent review, a prior assessment of the effects of growth-promoting substrates should be investigated before combining non-growth and growth-promoting substrates in a culture [2]. However, evidence indicates that the inhibition of degradation in the context of cometabolism generally occurs when a single bacterial strain is used as inoculum [2]. Thus, inoculum consisting of a mixture of bacterial strains should be considered so as to minimize the possible negative effect of growth-promoting substrates, a strategy supporting the use of bioaugmentation as discussed in Section 4.2.1.

## 5. Concluding Remarks

It is well established that increasing temperature leads to better degradation of pollutants. Thus, the use of thermophilic microbes provides an effective strategy to improve the overall biodegradation process. Yet, so far, most of the work on biodegradation has been carried out using mesophiles, and little is known about thermophiles.

This literature review has highlighted important findings on PHDT. For instance, more than 60% of PHDT strains isolated so far belong to one single genus, *Geobacillus*, while the family *Bacillaceae* represents more than 80% of strains. Some PHDT express a specific n-alkane hydroxylase gene, *LadA*, which has not been reported in mesophiles so far. In the degradation of monoaromatic compounds, evidence indicates that PHDT preferentially uses meta-cleavage pathways, yet both ortho- and meta-pathways exist in mesophiles. All of these observations show that thermophiles in general and PHDT in particular are endowed with unique biochemical pathways, and this is in line with the fact that thermophiles have been used as a source of enzymes or molecules in industrial biotechnology.

This review has also highlighted important gaps in our understating of PHDT. For instance, up to the present time, all PHDT described so far belong to the group of moderate- and extreme-thermophiles, and not a single hyper-thermophile has been reported so far. However, the exploitation of the genome information and the functional metagenomic (which bypasses the initial limitation of the in vitro culture) provide a great opportunity to discover and isolate novel PHDT strains (including hyper-thermophile).

Various strategies, such as bioaugmentation, cometabolism, and the new emerging genetically modified microorganisms and nanotechnology [94,95,96,97], have been tested to improve the efficiency of pollutant degradation by mesophiles, and some of these strategies have been used in real conditions of bioremediation. Thus, using the vast amount of information on mesophiles, experiments can now be scaled up for the bioremediation of contaminated samples by thermophiles.

## Figures and Tables

**Figure 1 ijerph-15-02782-f001:**
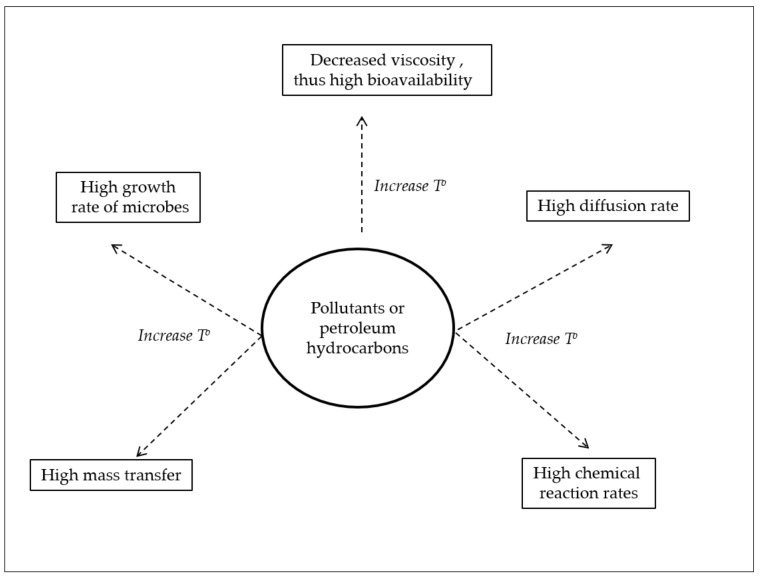
Various advantages of carrying out biodegradation reactions at higher temperatures (T°), thus, in the presence of thermophiles.

**Figure 2 ijerph-15-02782-f002:**
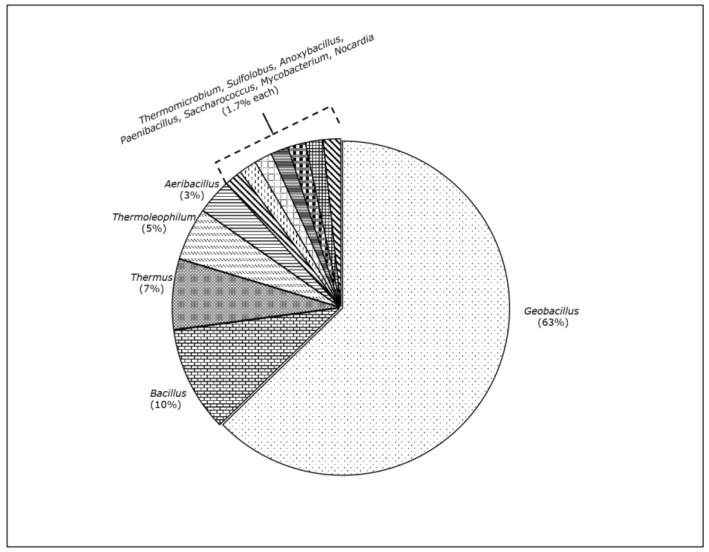
Distribution of genera of the 59 thermophilic strains degrading petroleum hydrocarbons summarized in Table 1.

**Figure 3 ijerph-15-02782-f003:**
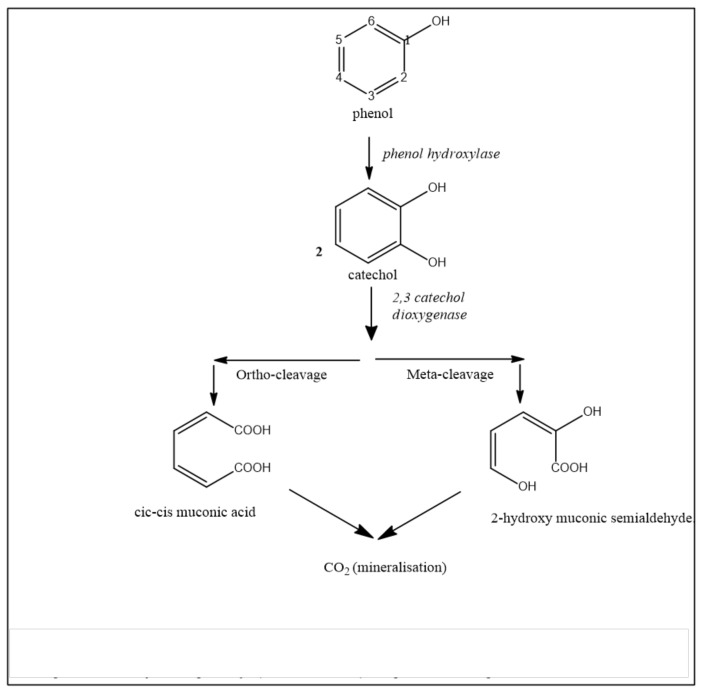
Pathways of phenol degradation. So far, only the meta-cleavage pathway has been described in thermophilic bacteria, yet both pathways (meta- and ortho-) are present in mesophilic bacteria.

**Figure 4 ijerph-15-02782-f004:**
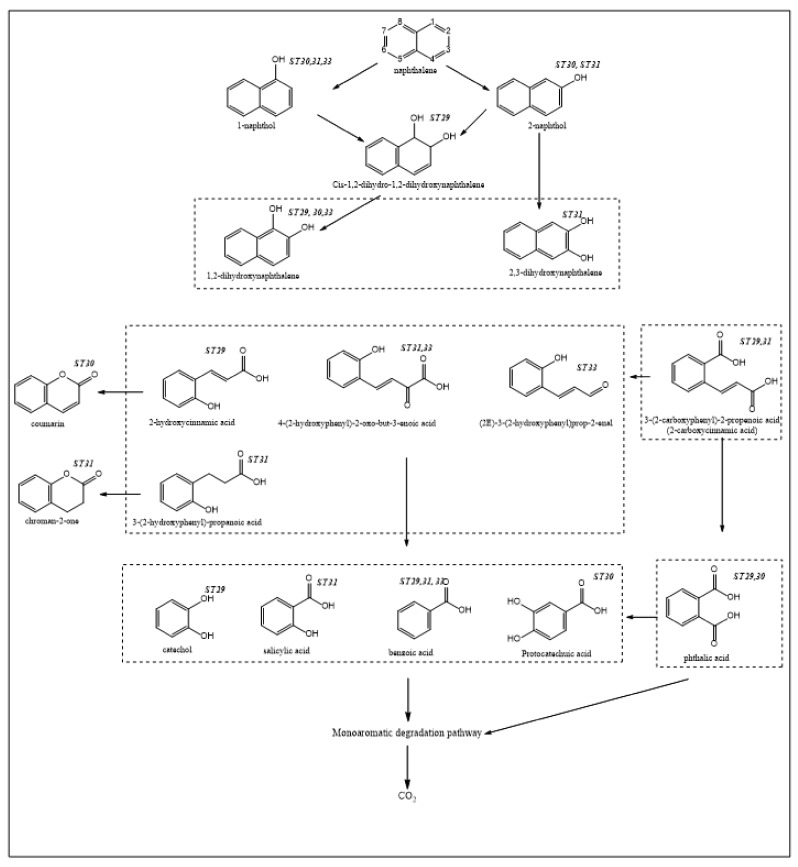
Possible pathways of naphthalene (NAPH) degradation by thermophilic microbes.

**Figure 5 ijerph-15-02782-f005:**
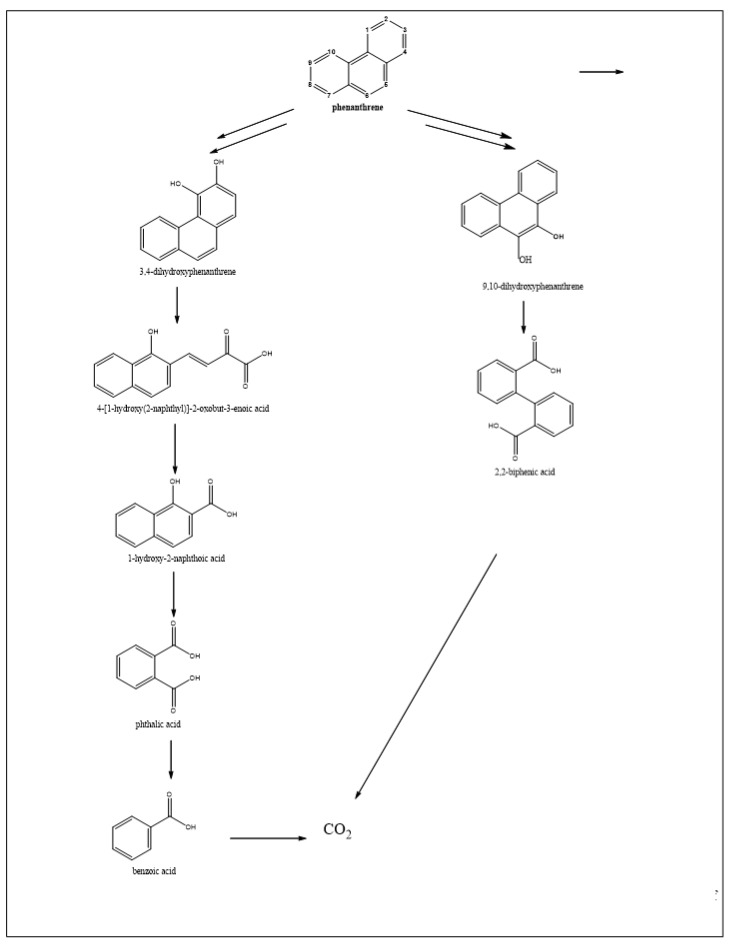
Biodegration pathways of phenanthrene in thermophilic bacteria. All these metabolites were identified from a strain of *Nocardia otitidiscaviarum* bacterium (ST29).

**Figure 6 ijerph-15-02782-f006:**
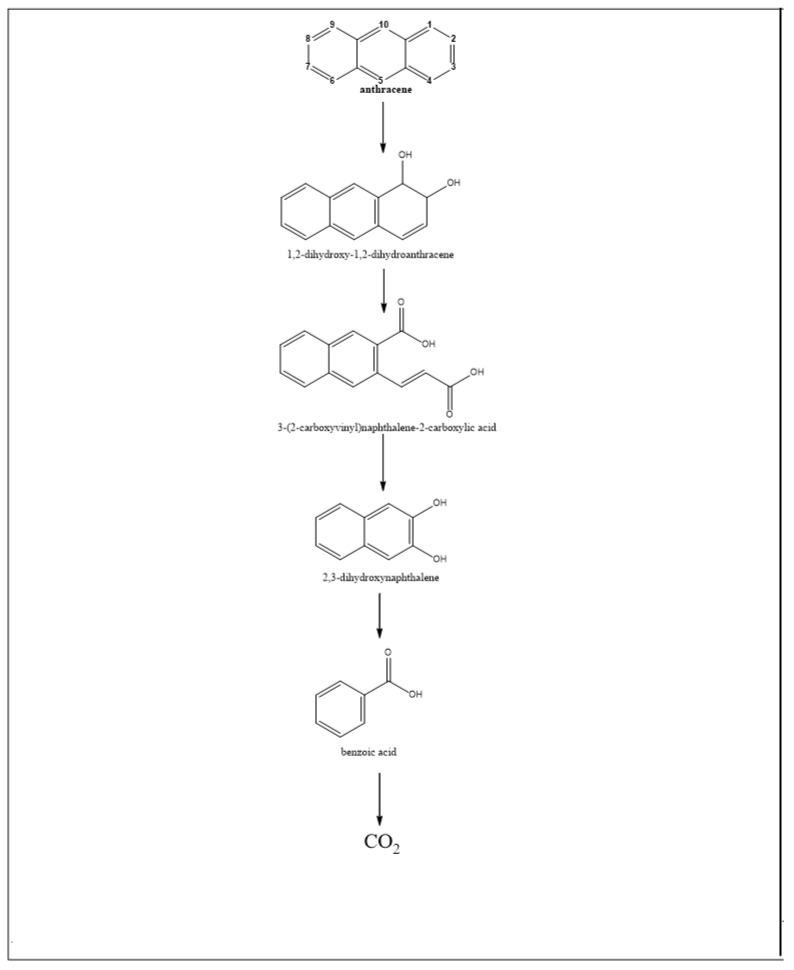
Biodegration pathways of anthracene in thermophilic bacteria. All these metabolites were identified from a strain of *Nocardia otitidiscaviarum* bacterium (ST29).

**Table 1 ijerph-15-02782-t001:** Summary of studies on biodegradation of petroleum products by thermophilic microbes.

	Study	Source of Samples	Conditions of Isolation	Thermophiles Used or Isolated	Range of Temperatures and/or Optimum Temperature (in Parentheses)	Petroleum Products Degraded or Other Characteristics of the Strains	Reference
n-alkanes	ST1	Samples of water or soil	Enrichment culture in the presence hydrocarbon at 57 °C	*Bacillus stearothermophilus*	(57 °C)	Hydrocarbons	[23]
	ST2	Mud sample from a littoral area of North Carolina, USA	Enrichment culture in the presence n-hexadecane at 50 °C	*Thermomicrobium fosteri* PTA-1,	42 to 70 °C (60 °C)	* n-alkanes (nC_10_–C_20_), 1-Alkenes (C_14_–C_1_), alcohol (C_12_–C_17_) and ketones (C_14_–C_17_) C_14_ * but not <C_9_ (n-alkanes), <C_9_ (alkene and ketones) and <C_11_ (alcohol)	[24]
	ST3	Mud samples from Yellowstone National Park, USA	Enrichment culture in the presence n-heptadecane at 60 °C	*Thermoleophilum minutum* YS-3 *Thermoleophilum album* YS-4	(60 °C)	* n-alkane C_10_–C_20_ but not those <C_9_ * C_6_–C_8_ alkenes, but not <C_6_ alkenes	[25,26]
	ST4	Hot water springs from Kunashir Island, Russia	Not provided	*Geobacillus stearothermophilus* 16 *Thermus ruber* 12	55–73 °C (60 °C)	Paraffins	[27]
	ST5	Mud and water from various parts in the USA	Enrichment in the presence of n-heptadecane at 60 °C	*T. album*	45 to 70° C (60 °C)	C_13_–C_30_	[28]
	ST6	Mud and water from the USA	Enrichment in the presence of n-heptadecane at 60 °C	*Geobacillus thermoleovorans*	55–65 °C	n-alkane C_13_–C_20_ but not those <C_9_ or >C_20_	[29]
	ST7	Contaminated samples from Kuwait	Enrichment in the presence of crude oil at 60 °C	*G. stearothermophilus* strains *KTCC-B7S* and *KTCC-B2M*	60 °C	C_15_–C_18_ n-alkanes but not of lower carbon numbers, alkenes and aromatics	[30]
	**ST8**	Production water samples from deep petroleum reservoirs, Japan	No enrichment	*G. thermoleovorans* B23 and H41	50–80 °C (70 °C for B23) (65 °C for H41)	* C_13_–C_26_ n-alkanes but not for C_9_–C_12_ * Evidence of terminal oxidation of n-alkanes	[31]
	ST9	Deep subterranean oil reservoir, China	Enrichment in the presence of crude oil at 73 °C	*Geobacillus thermodenitrificans* NG80-2	45–73 °C (65 °C)	* C_15_–C_36_ n-alkanes but not the short-chain (C_8_–C_14_) and those longer than C_40_ * No *alkB* gene homologs found in NG80-2	[32]
	ST10	Soil samples, Ireland	Enrichment at 60 °C in the presence of n-hexadecane	*G. thermoleovorans 27*, *Geobacillus caldoxylosilyticus* 17, *Geobacillus pallidus 2*, *Geobacillus toebii 1*, *Geobacillus* sp.	(60 °C)	* n-hexadecane * Expression of *AlkB* gene induced by n-hexadecane	[34]
	ST11	Samples from wells in the Dongxin reservoir (temperature 60–80 °C), China	Enriched culture in the presence of n-hexadecane	Thermophilic strain TH-2	40–85 °C (70 °C)	Degradation of the petroleum hydrocarbons, mainly n-alkanes	[35]
	ST12	Formation water from the Dagang oilfield and from a thermal spring in the Baikal rift zone, Russia	Not explained	*G. toebii* B-1024, *Geobacillus* sp. 1017, *Aeribacillus pallidus* 8m3	38–70 °C (60 °C)	* C_10_–C_30_ n-alkanes (for B-1024) * C_13_–C_19_ n-alkanes (for 1017) * C_11_–C_29_ n-alkanes (for 8m3) * Simultaneous presence of *alkB* and *ladA* genes involved in the degradation of alkanes	[36]
	ST13	Samples from active volcano of Santorini, Greece,	Bacteria were identified by *alkJ* probe, following culture in rich medium at 60–80 °C	*G. thermoleovorans* (4 strains) *G. stearothermophilus* (3 strains) *Geobacillus anatolicus (2 strains)* *Bacillus aeolius* (1 strain)	60–80 °C	* Long chain alkanes of crude oil * Use of *alkJ* probe to identify and select petroleum hydrocarbon degrading thermophiles (PHDT)	[38]
Monoaromatics	ST14	Industrial sediment, UK	Enrichment in the presence of phenol, at 55 °C	*G. stearothermophilus* PH24	(50 °C)	* Phenol and catechol * Cleavage of catechol at the meta-position	[39]
	ST15	Same as in ST13	Same as in ST13	Same strain as in ST13 (*G. stearothermophilus* PH24)	Same as in ST13	o-cresol, m-cresol, or p-cresol, 3-methylcatechol and 4-methylcatechol	[40,41]
	ST16	Soil samples that had been pasteurized for 10 min at 80 °C	Enrichment in the presence of m-cresol and phenol	*G. stearothermophilus* IC3	(50 °C)	* Phenol, m-cresol and benzoic acid * Phenol degradation via catechol, and cleavage via meta-position	[42]
	ST17	River sediment Tittabawassee, USA	Enrichment in the presence of phenol, at 55 °C	*G. stearothermophilus* BR219	(55 °C)	* Phenol * Characterization of phenol hydroxylase, the first enzyme in phenol degradation	[43,44]
	ST18	Not provided	Not provided	*G. stearothermophilus* FDTP-3	60–65 °C	* Phenol and catechol but no benzoic acid * catechol 2,3-dioxygenase gene was also characterized	[45]
	ST19	Water and mud from a hot spring, Iceland	Enrichment in the presence of phenol at 70 °C	*G. thermoleovorans* A2	(65 °C)	* Phenol and o-cresol, m-cresol, p-cresol, * meta-cleavage of phenol	[46,47]
	ST20	Not provided	Not provided	*Thermus aquaticus* ATCC25104 *Thermus* sp. ATCC 27978	(70 °C) (60 °C)	* Degradation of BTEX (benzene, toluene, ethylbenzene, and xylenes) * ^I4^C-labeled benzene and toluene were metabolized to ^14^CO_2_	[48]
	ST21	Samples from a waste treatment plant (UK) and a phenol-contaminated industrial effluent (lake Zealand)	Enrichment in presence of phenol at 50 °C	*Bacillus* sp.	50–55 °C	* Phenol degradation * Detection of catechol 2,3-dioxygenase * Cleavage of aromatic via the meta-pathway * Catechol 2,3-dioxygenase was inactivated at high oxygen	[49]
	ST22	Production water from an oil field, Tunisia	Enrichment in the presence of vanillic acid	*Aeribacillus pallidus*, VP3	37–65 °C (55 °C)	* Degradation of various aromatic compounds including benzoic, p-hydroxybenzoic, protocatechuic acids * Strain was also halotolerant (can grow in the presence of 100 g L^−1^ of NaCl)	[50]
	ST23	Volcanic site at Pisciarelli Solfatara, Italy	Already preselected	*Sulfolobus solfataricus* P2	(80 °C)	Anaerobic degradation of phenol	[51,52]
Polyaromatics	ST24	Compost from a field in Okayama, Japan	Enrichment culture in the presence of biphenyl as sole source of carbon, at 60 °C	*Geobacillus* sp. JF8	(60 °C)	* Biphenyl, chlorobiphenyl, NAPH, benzoic acid and salicylic acid but not PHEN, ANTH, benzene and xylene * Related to *G. Stearothermophilus*	[53]
	ST25	PAH-contaminated soil collected from Hong Kong	Not provided	*Bacillus subtilis* BUM and *Mycobacterium vanbaalenii* BU42	(55 °C)	Both strains could utilize PHEN as sole source of carbon * None of the strains could utilize BZP as sole source of carbon * In the presence of PHEN, only the strain BUM could degrade BZP	[54]
	ST26	Hot springs, compost and industrialwastewater	Not provided	Preselected mixture of strains of *Geobacillus* spp. *Thermus* sp.	60–70 °C	* Degradation of acenaphthene FLT, PYR and BZP, in the presence of hexadecane	[55]
	ST27	Soil samples from contaminated areas in Kowait	Enrichment in the presence of a mixture of PAHs and heterocyclic polyaromatics	*Bacillus firmus, Bacillus pallidus, Anoxybacillus* sp., *Paenibacillus* sp., *Saccharococcus* sp.	(60 °C)	* Degradation of NAPH, PHEN, ANTH, PYR and FLR, benzothiophene, dibenzothiophene dibenzofuran and carbazole * Inhibition of degradation of PAHs and heterocyclic compounds in the presence of glucose	[56]
	ST28	Compost sample	Incubation in the presence of PHEN at 60 °C	*Geobacillus* sp. (3 strains)	(60 °C)	PHEN, FLR and FLT degradation	[58]
	ST29	Petro-industrial wastewater soil, Iran	Selection in the presence of PHEN at 50 °C	*Nocardia otitidiscaviarum* TSH1	30–55 °C (50 °C)	* NAPH, PHEN and ANTH degradation * Evidence of meta-cleavage of NAPH * Detection of various metabolites of NAPH	[59,60]
	ST30	Geothermal oil field in Lithuania	Pre-culture in rich medium following screening in a culture in the presence of NAPH	*Geobacillus* sp. G27	(60 °C)	* Growth in the presence NAPH and ANTH, protocatechuic acid, benzene, phenol and benzene-1, 3-diol * No growth in the presence of catechol * Degradation of NAPH through protocatechuic acid via ortho-cleavage pathway	[61]
	ST31	Hydrocarbon contaminated compost, Germany	Enrichment in the presence of NAPH, at 60 °C	*G. thermoleovorans* Hamburg 2	(60 °C)	* Degradation of NAPH * Detection of phthalic and benzoic acid as NAPH metabolites	[62]
	ST32	Contaminated soilsamples from Yumen oilfield, China	Enrichment in the presence of PHEN, FLR and crude oil at 60 °C	*Geobacillus pallidus* XS2 and XS3	(60 °C)	* PHE, FLR and crude oil * Preference of nC_8_–C_19_ for XS2 strain and n-C_20_–C_38_ for XS3 strain * Production of emulsifier by the 2 strains	[63]
	ST33	Soil from a deep oil well, China	Enrichment in the presence of crude oil at 70 °C	*Geobacillus* sp. SH-1	45–80 °C (60 °C)	* Degradation of NAPH and nC_12_–C_23_ (from crude oil) * Detection of NAPH metabolites	[64]
	ST34	Produced water from Dagang petroleum reservoir at 73 °C, China	Already growing in petroleum environment	*G. stearothermophilus* strain A-2	40–75 °C (60 °C)	* NAPH, methylated-PHEN, FLR, benzo[*b*]fluorenes and long-chain alkanes (C_22_–C_33_) * Low degradation of shorter chains (C_14_–C_21_) * Production of bio-emulsifier	[65]
	ST35	Hot spring from Guerrero State, Mexico	Non explained	*Bacillus licheniformis* M2-7	(50 °C)	* BZP * Identification of phthalic acid as a BZP metabolite * Characterization of catechol dioxygenase enzyme	[66]

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
