# Peer review of "Current Status of the Degradation of Aliphatic and Aromatic Petroleum Hydrocarbons by Thermophilic Microbes and Future Perspectives"

_ijerph, 2018, doi:10.3390/ijerph15122782_

Reviewer 1 Report

Minor corrections (attached)

Author Response

I wish to thank the reviewer for his commments on the language. I have made all the corrections, and they are highlighted in yellow. 

Reviewer 2 Report

The reviewed manuscript "Current Status of the Degradation of Aliphatic and Aromatic Petroleum Hydrocarbons by Thermophilic Microbes and Future Perspectives" contains descriptions in the form of a review of the scientific literature allowing to gather information on the state of knowledge on the possibility of biodegradation of environmental pollutants arising from petroleum products using thermophilic microorganisms. The subject taken up by the Author is not only interesting, but also very important due to the quite extensive formation of oil-related pollutants in the environment and the associated difficulties in their disposal. The analyzes presented in the manuscript were supported by the literature review based on knowledge contained in 95 items included in the literature list. In my opinion, this is sufficient. The complexity of the issues presented by the Author, despite detailed description of some topics, requires from the reader a lot of concentration and attention. Despite that, the descriptions are presented in a clear way. The introduction at the beginning of the manuscript should in the final part justify its purpose, which should be its guiding idea. Its lack blurs the meaning of the analyzes in the text. Table 1 contains a very rich documentary material, which was used by the Author to make many references in the text. In my opinion, Chapter 4 is very extensive. Some of the its subchapters contain several-sentence comments that do not contribute anything. Rewording this section by reducing the amount of subsections can improve clarity and increase its scientific value. The last 5th Chapter entitled "Concluding Remarks and Future Perspectives" suggests that it contains a summary of prospects, as well as the conditions that accompany them. Unfortunately, the Author has included in this chapter content that is a repetition of the content discussed earlier. It seems that a better solution would be to title this chapter as a "Summary and conclusions" presented in the form of points, with the opinion of the Author, which methods and conditions accompanying them are developmentally perspective. The lack of such suggestions, despite the interesting content of the manuscript creates the impression that it is unfinished, and it is of great importance, because the Author's opinion would help other researchers to direct their research and significantly increase the scientific value of the paper. I believe that taking these remarks into account by the Author would allow the reviewer to fully recommend the manuscript for further publishing process.

Author Response

I wish to express my grateful to the reviewer for setting aside time for reading and commenting on the manuscript.  The reviewer has made 2 main comments, which I have addressed

1.    The first comment is about the reducing the size of the Chapter 4.

This chapter describes the limitation, the gap and, more importantly, work that needs to be carried out to improve our understanding of thermophiles. However, after careful consideration, I  have concurred with the reviewer to reduce it, by removing  the sections  on  “Genetically modified organisms and nanotechnology”, because , in fact, these 2 topics  have not been very fully studied yet  in mesophilic bacteria. However,  I have  mentioned them in the conclusion, and have quoted the appropriate references. ,

2.    The reviewer suggested that the Title of Chapter 5 be changed and  this section  be presented as  a“bullet-points”.  I concur with the reviewer to change the title, and, I propose to make shorther as a “conclusion” only.

A conclusion can be written in different styles, and I would prefer to present it as a text, instead of  as a  “bullet-point”. In addition, I have made some changes to limit this conclusion to main points.  

Reviewer 3 Report

The manuscript tilted “Current status of the degradation of aliphatic and aromatic petroleum hydrocarbons by thermophilic microbes and future perspectives” reviews the biodegradation of petroleum hydrocarbons by thermophiles. Some gaps that could limit the understanding of the activity of these microbes are discussed.

The manuscript makes mainly a compendium of the information available in the literature. The review of this information seems to me important. Taking in account that the information on the biodegradation mechanism in thermophiles is scarce, more critical review on this important aspect would be desirable. New techniques, such as metagenomics and new cultivation methods for non-cultivable microorganisms are not deeply discussed.

The authors claim that thermophiles do not express CYP (lines 252-254). This is not accurate. I suggest seeing the following references:

J. Inorg. Biochem. 2001, 91, 491-501.

J. Biol. Chem. 2003, 278, 608-616.

Biochem. Biophys. Res. Comm. 2005, 338, 437-445.

Appl. Microbiol. Biotechnol. 2011, 89, 1475–1485 .

Author Response

I thank the reviewer for  reading and commenting on the manuscript. The reviewer has made 2 main points regarding metagenomics and CYP, and below is my response.

1.                Metagenomics

In section 4.1.2, (L343), we did discuss  genomic approaches, and the use of the metabogenomics is in line this topic, thus we have added  a section on metagenomics,  and have quoted some of most recent and relevant publications on this metagenomic topics. 

I  have also mentioned this in the conclusion section.

2.                CYP presence in thermophiles

I agree with the reviewer that  thermophiles do express CYP. However, in the  manuscript,  I am referring to thermopiles that degrade n-alkanes only, and  so far (to the best of my knowledge),  there is no report indicating the expression of CYP in these thermophiles.

I have checked the suggested references, and none of them is related to the expression of CYP in the context of  n-alkanes degradation by thermophiles.

In fact, I did clearly mention this  in the text, in Page 13, line 245-246

I wrote the following   “Interestingly, so far, the two systems, CYP153 and AlmA, have not been reported in thermophilic bacteria degrading n-alkanes”..

However, in light of the reviewer comments, and  to avoid any confusion, I have re-written the sentence in line 252-254, and have made it clear that  only  n-alkane degrading thermophiles do not have CYPs.

So, this section reads now as follows (line 252-254):

“in summary, so far, research has shown that n-alkane degrading mesophiles and thermophiles express AlkB monooxygenase enzymes; however, in addition, these mesophiles have CYP153 and AlmA enzymes, while these thermophiles express LadA monooxygenases..”